# Application of Spectral Index-Based Logistic Regression to Detect Inland Water in the South Caucasus

James Worden, Kirsten M. de Beurs * , Jennifer Koch and Braden C. Owsley

Department of Geography and Environmental Sustainability, University of Oklahoma, Norman, OK 73072, USA; james.a.worden-1@ou.edu (J.W.); jakoch@ou.edu (J.K.); braden.owsley@ou.edu (B.C.O.)
* Correspondence: kdebeurs@ou.edu

**Abstract:** The Caucasus is a diverse region with many climate zones that range from subtropical lowlands to mountainous alpine areas. The region is marked by irrigated croplands fed by irrigation canals, heavily vegetated wetlands, lakes, and reservoirs. In this study, we demonstrate the development of an improved surface water map based on a global water dataset to get a better understanding of the spatial distribution of small water bodies. First, we used the global water product from the European Commission Joint Research Center (JRC) to generate training data points by stratified random sampling. Next, we applied the optimal probability cut-off logistic regression model to develop surface water datasets for the entire Caucasus region, covering 19 Landsat tiles from May to October 2019. Finally, we used 6745 manually classified points (3261 non-water, 3484 water) to validate both the newly developed water dataset and the JRC global surface water dataset using an estimated proportion of area error matrix to evaluate accuracy. Our approach produced surface water extent maps with higher accuracy (89.2%) and detected 392 km$^2$ more water than the global product (86.7% accuracy). We demonstrate that the newly developed method enables surface water detection of small ponds and lakes, flooded agricultural fields, and narrow irrigation channels, which are particularly important for mosquito-borne diseases.

**Keywords:** water index; logistic regression; Landsat; Caucasus; global land cover datasets

## 1. Introduction

Water is arguably the most essential compound related to carbon-based life [1]. However, our relationship with water can change with the quantity present in a system. Too much water—flooding—can cause loss of life and disease prevalence, while not enough water—drought—can cause famine and dehydration [2]. In addition, there is a relationship between increased surface water and the abundance of mosquitoes [3]. An increase in mosquito breeding areas may contribute to an outbreak of malaria in regions that are prone to such outbreaks, making it worthwhile to detect and quantify surface water in those regions. Due to limitations in their flight range and survival rate, mosquitoes are restricted to areas that contain persistent pools of water [4,5]. Human interactions with mosquitoes are most likely to occur in proximity to mosquito habitats [6]. The South Caucasus is not typically an area that comes to mind when thinking about malaria outbreaks; however, after early eradication of malaria in the mid-1900s, this region saw a malaria resurgence in the mid-1990s to the early 2000s. To better understand which regions have the environmental potential for another resurgence, we are interested in studying surface water, including smaller water bodies such as irrigation channels and flooded croplands.

With satellite image availability starting in the 1970s, monitoring the entire water cycle has been an important area of research [7]. For example, several evapotranspiration products have been developed using satellite data, such as that from NASA's Moderate Resolution Imagining Spectroradiometer (MODIS) [8] and the ESA's Medium Resolution Imaging Spectrometer (MERIS) and Advanced Along-Track Scanning Radiometer (AATSR) [9]. These datasets have subsequently been used and applied in a variety of

research, for example, to predict evapotranspiration over the Nile Delta Region [10]. Significant efforts have also been made to use satellite data to better understand water quality. Parameters such as turbidity and chlorophyll-a have long been studied [11,12]. With the advent of increased computing power and progressively available, highly accurate, remotely sensed data, machine learning methods are increasingly applied for water quality monitoring [13,14].

In this paper we are particularly interested in the detection of surface water. Many studies around the world have applied Landsat imagery in the detection of surface water dynamics [15–19]. As a result of the importance of surface water maps, as well as the large spatio-temporal variability of water bodies, many ways to detect the location and amount of surface water from space have been developed [20]. Early research in surface water detection typically applied a simple threshold to single band images, for example, initially to the near infrared band from Landsat MSS [21], and for subsequent Landsat sensors to the shortwave infrared band [22]. Others used density slicing, a slightly more complicated single band method, for example to identify water lines in tidal flats in South Korea [23]. Density slicing using a normalized threshold to allow for varying environments has also been applied to active remote sensing data to identify Canadian Prairie Potholes [24]. In other research using RADAR data, normalized thresholds were combined with image segmentation to detect surface water in Canadian wetlands [25]. However, when using optical satellite data research has shown that multiband spectral indices are better at detecting land surface water than single spectral bands [26]. As a result, a very large number of water indices have been developed [20], with the Normalized Difference Water Index (NDWI; [27]), Modified NDWI (MNDWI; [28]), Automated Water Extraction Index (AWEI; [29]), and Enhanced Water Index (EWI; [30]) some of the most frequently used indices. Since the different spectral indices perform differently depending on regional characteristics, we have previously evaluated four different water indices (NDWI, MNDWI, AWEIsh, AWEInsh) and a water detection method based on EVI, NDVI and MNWDI [31] for three study regions in the South Caucasus [32]. We found that while all of the evaluated indices were relatively accurate, the MNDWI index resulted in the most accurate open surface water maps for three regions in the South Caucasus. To derive a water/non-water map from spectral water indices, threshold methods are often applied. However, because of variations in the physical environment over space and time, it is often not straightforward to establish a constant threshold value [32]. Some authors have resorted to the evaluation of multiple threshold values to determine the most optimal threshold for a specific region [33]. For example, Jiang et al. (2014) evaluated a series of threshold values to distinguish water pixels from mountains and urban areas. Others used a confusion matrix to determine an optimal threshold value [28]. Previously, we have argued that it is possible and advantageous to use an iterative process to determine the optimal probability cut-off for each individual image [32]. We apply that same flexible methodology in this paper.

Regardless of the type of data being used or the type of method being applied to identify surface water, it is well known that training data quality can significantly impact the accuracy and effectiveness of classification models [34,35]. In the past it was common to argue that training data should ideally be derived from in situ data [36]. However, over time the importance of large numbers of training samples has become increasingly clear [34]. Imbalanced training data due to rare land cover classes such as water is another common problem in remote sensing classification studies [37]. To solve this imbalance problem, some researchers have down-sampled majority classes [38], while others have given rare training observations higher weights [36]. Obtaining quality training data can be a consumptive process, and cost, time, and processing power are common barriers [39]. In the early 2000s, the Landsat images themselves, combined with a priori knowledge of the study area, were sometimes used to create training data in different land cover classifiers as an alternative to in situ training [40]. In our earlier study we used very high-resolution images from Google Earth as training data, which is a relatively common approach [32]. However, manual evaluation of training points can be a time-consuming and challenging

process, resulting in temporally static data that decreases in quality as time increases from the acquisition date. Some have used previously classified data as training data; for example, in one paper, the National Land Cover Database (NLCD) was used to classify MODIS data [41].

Global land cover datasets are increasingly becoming freely available. Besides general land cover datasets such as the ESRI 2020 global land use/land cover map derived from ESA Sentinel-2 imagery [42] and the ESA WorldCover 10 m 2020 product (https://esa-worldcover.org/en, 24 November 2021), the European Commission's Joint Research Centre (JRC) has also developed specific land cover datasets such as the Global Man-made Impervious Surface (GMIS) [28] and Global Surface Water [26]. The JRC Global Surface Water dataset is a regularly updated water dataset produced from the 30 m Landsat archive (1984 to the present) which is highly accurate in the detection of large water bodies but struggles to detect smaller water features or vegetated water in a flooded landscape [32,43]. While global products have important value in providing consistent data around the globe, it is unrealistic to expect these products to have consistent global accuracy. Data generated with a local or regional focus may consequently have higher accuracy for specific areas. Nevertheless, global products can be of enormous value. For example, past studies have shown that it is possible to improve upon existing global land cover products by using these global land cover datasets themselves to generate training samples for more advanced classifications [44,45]. Combining training samples from existing land cover products with other land cover classification techniques such as random forests (RF) has led to improvements in accuracy for detecting vegetation compared to the original dataset [46].

For this study, we trained a logistic regression water model based on the Modified Normalized Difference Water Index (MNDWI) for the Caucasus region using the JRC Global Surface Water product to establish training points. We improved the detective capability of the band ratio water index by establishing a relationship between water's spectral signature as captured in the index value and a threshold, selected using a logistic regression model and the optimal probability cut-off (OPC) method. For each probability map generated by the model we applied the optimal threshold to produce the most accurate map of water/non-water. We applied the OPC method to data for the entire South Caucasus region covering the period from May to October 2019. We selected this period for two reasons: first, the period between May and October is mainly snow-free for most of the South Caucasus; second, we selected 2019 because it mostly matches the validation samples collected from the very high-resolution satellite images. In summary, we show that it is possible to improve water mapping capabilities, especially for smaller water bodies by taking advantage of existing land cover detection datasets to train and classify more detailed surface water maps.

## 2. Background and Study Region

The South Caucasus region is made up of three countries: Armenia, Azerbaijan, and Georgia. This region hosts many different climate zones ranging from alpine mountains to subtropical lowland plains [47–49]. From the 1800s, malaria has been prevalent in the Caucasus Region, with 600,000 cases recorded in Azerbaijan in 1934 [50]. By the 1950s, the annual number of documented malaria cases reached a high of 781,239 [51]. After a comprehensive effort from the Global Malaria Eradication Campaign in the 1960s, malaria incidents declined, and two malaria species were eradicated. A third malaria species (*P. vivax*) escaped elimination and precipitated a surge of malaria cases following the collapse of the Soviet Union [52]. Malaria even flared up as far north as Moscow between 1999 and 2008 [53]. The South Caucasus experienced significant land reform and privatization of the agricultural sector in the 1990s, which led to the segmentation of large agricultural plots into smaller private and commercially owned farms. This process, which left irrigation systems degraded in some agricultural areas [47–49], combined with post-Soviet conflicts in the region (most notably between Armenia and Azerbaijan) to create the ideal conditions for the resurgence of malaria that occurred from the mid-1990s to the

early 2000s. The extensive wetland areas in the Caucasus form ideal mosquito breeding grounds for the *Anopheles* mosquito, which can carry the *P. vivax* species of malaria. The *Anopheles* mosquito requires persistent breeding pools caused by intraseasonal rainfall to proliferate. In addition, the warm and humid summers in this region allow for the maturation of more than one generation of *P. vivax* sporozoites per year [53].

## 3. Data and Methods

### 3.1. Landsat

We used Landsat Collection 1 Surface Reflectance data collected by the Landsat 8 Operational Land Imager (OLI). This is a Level-2 Science Product, atmospherically corrected using the Land Surface Reflectance Code (LaSRC), with a spatial resolution of 30 m. We included all images available with less than 30% cloud cover for the South Caucasus region, consisting of 19 WRS path/rows (Figure 1), for the period from May to October 2019. We selected images between May and October to avoid extensive periods with snow cover, which was an issue in higher elevations especially. In addition, we selected the images from 2019 because it matched the year for most of the very high-resolution images used for our validation point collection, which was carried out primarily in 2019, with a few more points collected in 2020. We selected 30% as our cloud cover cut-off because it resulted in a substantial number of images per path/row; when we lowered the cut-off we found far fewer images, making the final classification substantially less accurate. In addition, we found that if we increased our cloud cover cut-off, clouds became a significant problem, and occasional masking issues where clouds were missed resulted in spurious water detection. The images include a quality assessment band generated using the CFMask algorithm [54], which we used to filter cloudy or otherwise corrupted data.

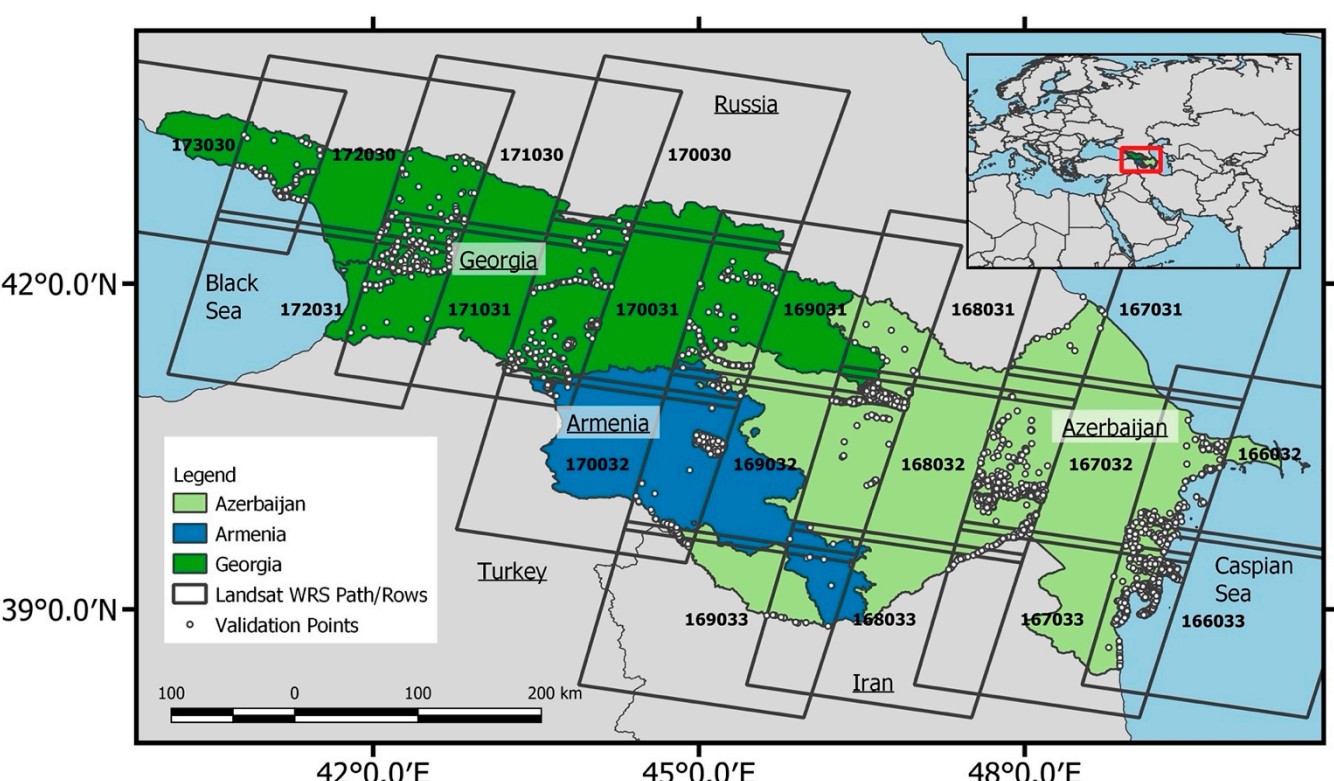

**Figure 1.** 19 WRS Path/Row tiles covering Armenia, Azerbaijan, and Georgia in the South Caucasus region, with validation points located in the areas where tiles overlap.

The 19 Landsat tiles are spread across eight paths (166–173; Figure 1), and each group of images in a path is processed as a set (for example, path 167 is a set of three path/rows: 167031, 167032, and 167033).

We calculated the Modified Normalized Difference Water Index (MNDWI; Equation (1), [28]) for each Landsat dataset to train the logistic regression model using sampled points. In a previous study [32], we evaluated NDWI, MNDWI, AWEIsh, AWEInsh, and a water classification algorithm based on NDVI/EVI and MNDWI to find surface water for three different study regions in the South Caucasus. We found that both the MNDWI and AWEInsh indices resulted in one of the highest overall accuracies, and one of the lowest levels of water underestimation. However, MNDWI performed slightly better in mountainous areas. As a result, we have selected MNDWI as our index of choice for this study. MNDWI is calculated as follows:

$$MNDWI = \frac{(Green - SWIR_1)}{(Green + SWIR_1)} \tag{1}$$

### 3.2. JRC Water Training Data

The JRC Global Surface Water is a regularly updated water dataset produced from the 30 m Landsat archive (1984 to the present) [26]. The dataset was created by applying an expert system classifier that segregates pixels into one of three target classes: water, land, and non-valid. While the water products offered by the JRC are highly accurate in the detection of large water bodies, they struggle in detecting smaller water features or vegetated water that results from a flooded landscape [32,43].

We used the JRC Monthly Water History dataset in this study to establish our training points. We developed the training points as follows: First, for each set of Landsat images meeting our criteria (those on a given path and date), we selected the corresponding JRC Monthly Water History map. We then selected a stratified random sample of 1500 points (750 water, 750 non-water) based on the JRC water map for each Landsat path.

### 3.3. Validation Dataset

Within the country boundaries of Georgia, Armenia, and Azerbaijan, we independently and randomly selected and then evaluated and classified 6491 stratified validation points (3238 non-water and 3253 water) using Google Earth imagery. The validation points were collected in 2019 and 2020, and we used the highest resolution basemap available on Google Earth. The basemap imagery was entirely from Maxar Technologies, Centre national d'études spatiales (CNES) and Airbus. In other words, the imagery for the basemap came from the following very high-resolution satellites: Worldview series and Quickbird (Maxar), Pleiades (CNES), and SPOT 6/7 (1.5 m; Airbus). As a result, the basemap served on Google Earth which was used for validation had a spatial resolution of 70 cm or less for most places, with a maximum spatial resolution of 1.5 m. We limited the validation point locations to the Landsat paths' overlapping sections in order to increase the number of uses in validating the model across the South Caucasus region (Figure 1), as each point can be used twice to validate the logistic regression model in adjacent paths. Although theoretically the highest quality data are in the center of the image, this gives us a more conservative estimate of the accuracy of our method.

Based on the three study regions in the South Caucasus studied in [32], we estimated that surface water occupied about 0.9% of the land surface in the South Caucasus (note that in this paper focusing on the entire South Caucasus, we find that water occupied just over 2%). Since water is a relatively uncommon land cover class in the study area, it is rare to randomly select a pixel where JRC water is missed. For example, if for simplicity we assumed that water occupies exactly 1% of the entire landscape, we would need to sample 100 random points to find one sample point with water. This means that if we were interested in finding 250 sample points with water, we would have to sample $250 \times 100 = 25{,}000$ random points. Our previous study ([32]; Table 5 in that paper) found that the JRC data underestimates the amount of water in three study regions in the South

Caucasus by about 15.7%. In other words, if we would sample 25,000 random points, statistically we would find approximately 250 points with water (assuming water occupies 1% of the landscape), and out of those 250 points, only 39 (15.7%) would be water data that was not found in the JRC dataset. Considering these low numbers, even when we sample 25,000 points, it seemed that we would need to apply a different strategy to evaluate water that was missing in the JRC dataset. Based on the results in [32], we estimated that our OPC method underestimates water by about 5.2%. This means that we estimate that we find about 10.2% more water in the OPC data than in the JRC data. Therefore, to offset this bias, we selected the 500 water validation points focused on areas where JRC missed water, but OPC did not; in other words, focusing on the 10.2% discussed earlier.

The 500 additional points were also manually classified using Google Earth imagery, separating the points into four classes: water, not water, cloudy, and water fraction (WF). We labeled pixels as WF if there was subpixel water present (water fraction less than 50%). Pixels classified as cloudy or WF were omitted from the additional water validation points, leaving 254 points added to the validation dataset. Most pixels dropped were because of a water fraction, that is, there was water in the grid cell, but it occupied less than 50%. Including the post-classification validation points, the total number of the validation sample size rose to 6745 points (3261 non-water, 3484 water). In other words, these 254 points occupied about 7.5% of our total water sample. We then used the validation dataset to determine the performance metrics of overall accuracy, sensitivity, specificity, ROC, and concordance for the OPC and JRC Max Extent water maps.

### 3.4. Generating Optimal Probability Cut-Off Water Maps

The previously developed method [32] depended heavily on both hand-selected training and validation points. Here, we scaled up this methodology and applied the OPC method to all Landsat images covering the South Caucasus region from May to October 2019. Instead of manually identifying thousands of training points, we used training data sampled directly from the JRC Monthly Water History. We believe that this is a valid method because previously we found that the JRC data had a much higher user's accuracy (95.7%) than producer's accuracy (84.7%) for three study regions in the South Caucasus [32]. In other words, we found a relatively high error of omission in the JRC data (15.7%), but a relatively low error of commission (4.3%). As a result, if the JRC data indicated water, we found water in the validation sample 95.7% of the time. This means that we can use the JRC data as a training dataset for our study, because while the JRC data misses 15.7% of the water, it rarely overestimates water. Figure 2 presents an overview of the method applied to each available Landsat image.

We used the JRC Monthly Water History product to generate logistic regression training points for each Landsat path. After generating the training points, MNDWI values were extracted from the point locations and used to train a logistic regression model, resulting in slope and intercept values for the water model. The slope and intercept values were then entered into the logistic regression probability equation:

$$p = \frac{1}{e^{-(a+bx)}} \tag{2}$$

In this equation, $p$ represents the probability that water is present in a pixel, $a$ is the y-intercept, $b$ represents the slope, and $x$ represents the MNDWI value. We used the probability Equation (2) to create water probability maps.

The optimal probability cut-off (OPC) method determines the threshold used to distinguish between water and non-water. The OPC is an iterative process based on the receiver operator characteristic (ROC), testing all probability cut-off values that produce the most significant degree of accuracy for water classification [55]. There is still potential for confusion between water pixels and other surfaces, such as barren mountains (often in shadow) and urban impervious surfaces. As part of the final water map development, we

used a digital elevation model to mask steep mountain slopes, and an impervious surface dataset to mask urban areas [32].

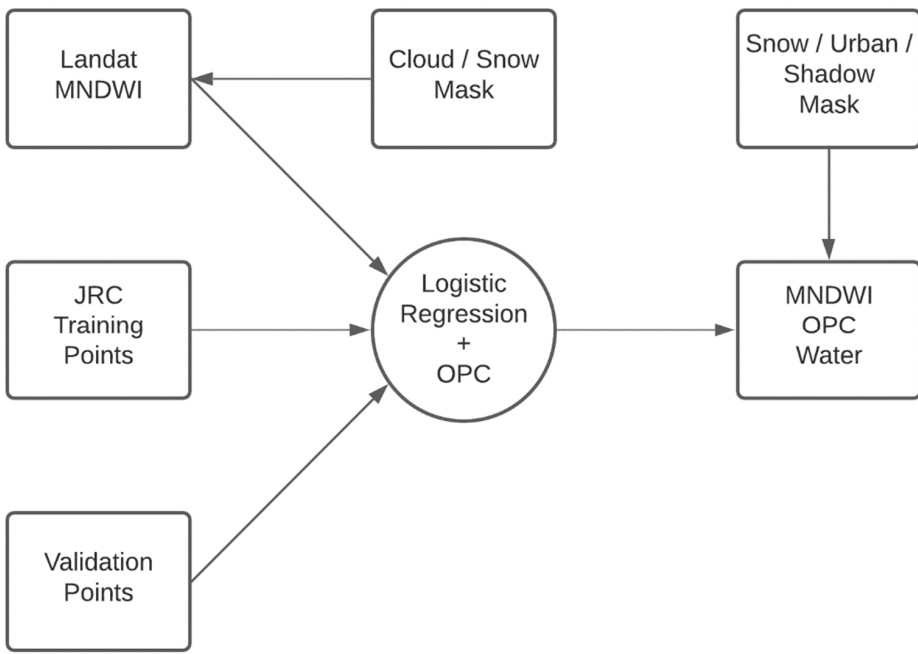

**Figure 2.** The process of generating OPC water maps from Landsat datasets using the JRC Monthly Water History for training and using Google Earth imagery to manually classify validation points. OPC: Optimal Probability Cut-off. ECJRC: European Commission's Joint Research Centre Global Surface Water. MNDWI: Modified Normalized Difference Water Index.

### 3.5. Performance Evaluation of Water Maps

The performance of the JRC and OPC water datasets was assessed by comparing the accuracy of the Max Extent for the entire Caucasus region. This included comparing the accuracy between the individual path/row sets of the JRC Monthly Water History and the OPC water maps. To evaluate the two datasets, we developed an error matrix of sample counts and an error matrix of estimated proportions. Olofsson et al. (2013) warn of calculating overall and producer's accuracy estimations directly from a sample count error matrix, suggesting that such a matrix would not account for the variation in estimation weights based on proportions of the mapped classes [56]. An error matrix describing the estimated proportion of area provides a more appropriate mechanism for evaluating the producer's and overall accuracies of land cover classification maps. Following this guidance, we calculated the overall accuracy from the sample count matrix and compared the result to the user's, producer's, and overall accuracies based on the mapped area class proportion [56]. We compared the performance of the JRC Monthly Water History and the OPC water datasets for each Landsat path in the Caucasus region. We used the sample count matrix for each observation to calculate overall accuracy. The producer's accuracy for the water class is derived from the estimated proportion of water in the map area.

## 4. Results

We used a random stratified sample of the annual JRC Max Extent water map to construct a dataset to train a logistic regression water model. The index-based logistic regression model uses the OPC method to classify water across Armenia, Azerbaijan, and Georgia from May to October 2019.

We compared the overall accuracy between the JRC Monthly Water History and the OPC water maps for each observation of the Landsat datasets. We then evaluated the data products with a traditional error matrix, including the overall estimated area of

surface water/land detected, and an error matrix using an unbiased estimator of the area proportion.

### 4.1. Overall Accuracy

To thoroughly assess the performance of land cover classification maps, we generated an error matrix [56]. The matrix displays the proportion of mapped area for each category (water/non-water), including the user's, producer's, and overall accuracies (Table 1). We found that the OPC method detects almost 400 km$^2$, or 10.5%, more water in six months of 2019 than the annual JRC Max Extent (4130 km$^2$ vs. 3738 km$^2$; Table 1). Calculating the overall accuracy of the two detection methods gives a result of 85.2% for the JRC Max Extent and 88.6% for the OPC Max Extent.

**Table 1.** Error matrix of sample-based validation points from the JRC Max Extent water map for 2019 and the OPC Max Extent water map from May to October 2019.

| JRC Jan–Dec 2019 | Non-Water | Water | Total | Overall Accuracy (%) | Mapped Area by Class (km$^2$) | Proportion of the Mapped Area by Class (Wi) |
|---|---|---|---|---|---|---|
| Non-Water | 2675 | 408 | 3083 | 85.2 | 182,060 | 0.9799 |
| Water | 586 | 3076 | 3662 | | 3738 | 0.0201 |
| Total | 3261 | 3484 | 6745 | | 185,798 | 1 |
| **OPC May–Oct 2019** | **Non-Water** | **Water** | **Total** | **Overall Accuracy (%)** | **Mapped Area by Class (km$^2$)** | **Proportion of the Mapped Area by Class (Wi)** |
| Non-Water | 2834 | 342 | 3176 | 88.6 | 181,668 | 0.9777 |
| Water | 427 | 3142 | 3569 | | 4130 | 0.0222 |
| Total | 3261 | 3484 | 6745 | | 185,798 | 1 |

Considering the error matrix using the estimated proportion of area, both datasets are highly accurate, retaining an overall accuracy of 86.7% (JRC) and 89.2% (OPC) (Table 2). The JRC and OPC water maps accurately detected the non-water land cover in the South Caucasus region, having a commission/omission error of 13%/0.4% (JRC) and 11%/0.3% (OPC). Examining the producer's accuracy for the water land cover class, we see the JRC dataset correctly detects 11.5% of the surface water, omitting a large proportion of water from the map. The OPC water map performs significantly better ($p < 0.01$), having a producer's accuracy of 15.5%, indicating a loss of performance from the underestimation of surface water area.

**Table 2.** The estimated proportion of area error matrix, including user's, producer's, and overall accuracies, between the JRC Max Extent water map for 2019 and the OPC Max Extent water map from May to October 2019.

| JRC Jan–Dec 2019 | Non-Water (-) | Water (-) | Total (-) | User's Accuracy (-) | Producers Accuracy (-) | Overall Accuracy (%) |
|---|---|---|---|---|---|---|
| Non-Water | 0.8502 | 0.1297 | 0.9799 | 0.8677 | 0.9962 | 86.7 |
| Water | 0.0032 | 0.0169 | 0.0201 | 0.8400 | 0.1153 | |
| Total | 0.8534 | 0.1466 | 1 | | | |
| **OPC May–Oct. 2019** | **Non-Water (-)** | **Water (-)** | **Total (-)** | **User's Accuracy (-)** | **Producers Accuracy (-)** | **Overall Accuracy (%)** |
| Non-Water | 0.8725 | 0.1053 | 0.9778 | 0.8923 | 0.9970 | 89.2 |
| Water | 0.0027 | 0.0196 | 0.0222 | 0.8804 | 0.1567 | |
| Total | 0.8751 | 0.1249 | 1 | | | |

### 4.2. Accuracy Assessment by Region

The previous section evaluated the JRC and OPC max extent water products' overall accuracy from all validation points across the South Caucasus region. However, because the spatial and temporal distribution of surface water is not uniform across the study region, it is also essential to assess the overall accuracy of the JRC and OPC methods for

each Landsat dataset in the study period. Figure 3 describes the overall accuracy between the two datasets across all Landsat paths from May to October 2019. With the western paths 171, 172, and 173 being the exception, we observe that the overall accuracy for both water detection methods remains high, ranging from 87% to 98%. The JRC and OPC max water extent share similar accuracy trends, each outperforming the other at various points for the selected months (Figure 3). The OPC and JRC accuracy dropped in the three westernmost Landsat paths (171, 172, 173), with a lower overall accuracy of 80%. These three paths cover the western half of Georgia and have the least amount of surface water area of the eight paths that comprise the Caucasus study region.

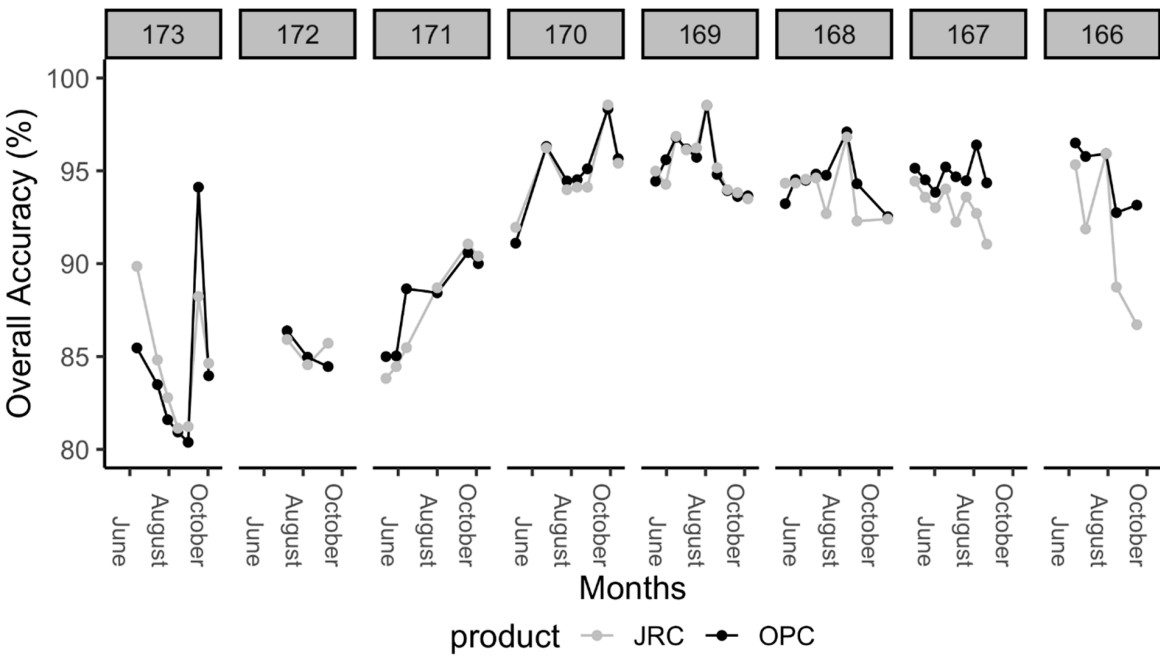

**Figure 3.** The overall accuracy between the JRC monthly water history and the OPC method from May to October 2019 for eight different Landsat paths. Note that the accuracy (y-axis) has a lower limit of 80%. The paths are visualized according to the map direction, with path 173 in the West (left) and path 166 in the East (right).

### 4.3. Producer's Accuracy

The high overall accuracy of the JRC and OPC Max Extent water datasets is partly due to the detection of non-water class, which is the dominant land cover class covering more than 97% of our study area. However, the main focus of these two water detection methods is to detect water. When we focus on just the producer's accuracy of detecting the water class, the performance is not as strong as the high values of overall accuracy.

Figure 4 gives insight into the water detection difficulties among the different Path/Row sets. The Landsat paths in the eastern part of our study area (esp., 166, 167, and 169) boast the highest producer's accuracy in detecting surface water, because they contain the largest and most stable water bodies from which to extract the MNDWI values for the model's training and validation points. There are some temporally unstable water bodies in paths 167 and 168 due to heavy vegetation growth in the larger water bodies, which can vary throughout the year. We believe that these water bodies cause a loss in performance in the JRC dataset compared with the OPC dataset [57]. We attribute the drop in performance in the western paths to the temporal instability of the braided river system. Visual inspection with high-resolution imagery revealed that these paths contain sinuous and non-sinuous braided river systems, with a few small reservoirs and lakes (Figure 5). Braided river systems can rapidly change due to seasonal flow regimes and sediment transport, causing changes in water location and discharge [58]. High-discharge events and water channel drift can influence the spectral signature and MNDWI values of a pixel by changing the

fraction of subordinate land cover classes within the satellite image's spatial resolution [33]. Variations in these river conditions can change the land cover type of a validation point, making it unreliable. The consequence of confining the validation points to the Landsat paths' overlapping areas is a condensed sampling of the river systems in these areas, resulting in a loss of overall accuracy.

### 4.4. Water Detection in the Caucasus

For our entire study region covering Azerbaijan, Armenia and Georgia, we found that the OPC method detected approximately 10.5% more water for 2019 than the annual JRC Max Extent dataset. This matches very closely the difference in underestimation we found in our previous paper for three focused study regions in the Caucasus [32], where we estimated that the JRC Max Extent omission error was 15.7%, compared with 5.2% for the OPC method. Most of the omitted areas are focused on very small bodies of water, such as irrigation channels, which are especially relevant for this study. The amount of additional water found with the OPC method was not equal for all three countries. Georgia showed the lowest amount of surface water, about 404 $km^2$ for the JRC method, and 444 $km^2$ for the OPC method. This means that we found about 10% more water in Georgia using the OPC method. The two methods estimated almost equal amounts of surface water for Armenia, 1395 $km^2$ (JRC) and 1415 $km^2$ (OPC); the OPC method only found about 1.5% more water. Azerbaijan revealed the largest amount of surface water, and we found that the OPC method estimated almost 17% more water in Azerbaijan (2266 $km^2$) than the JRC method. Many of these additional water bodies are smaller flooded fields and smaller irrigation channels.

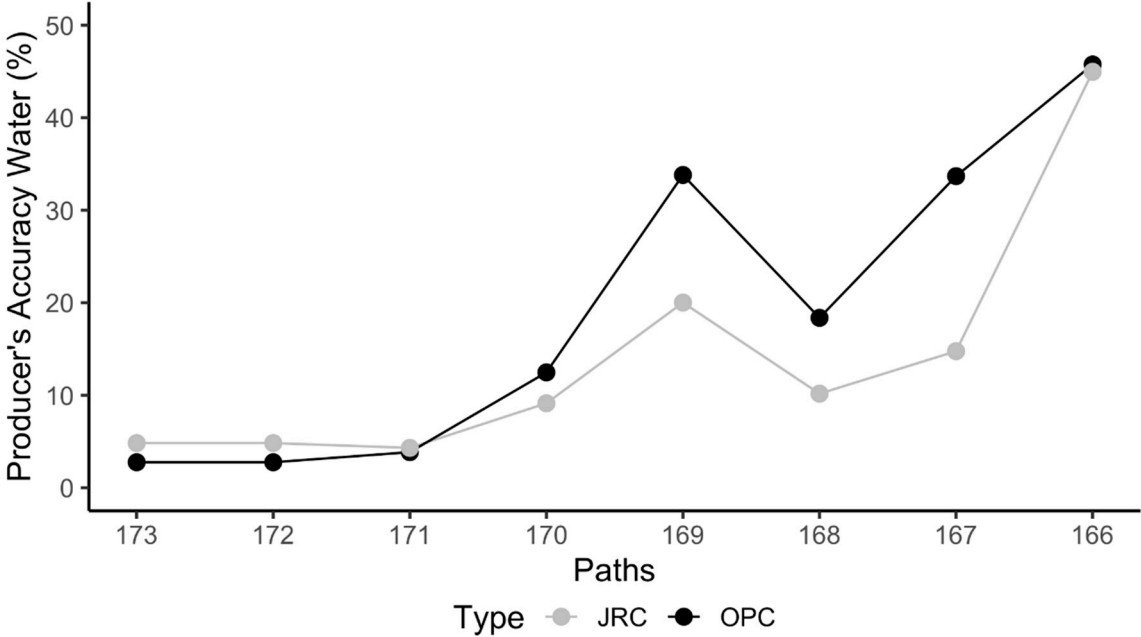

**Figure 4.** The JRC and OPC Max Extent water map producer's accuracy for detecting the Caucasus region's water class for 2019.

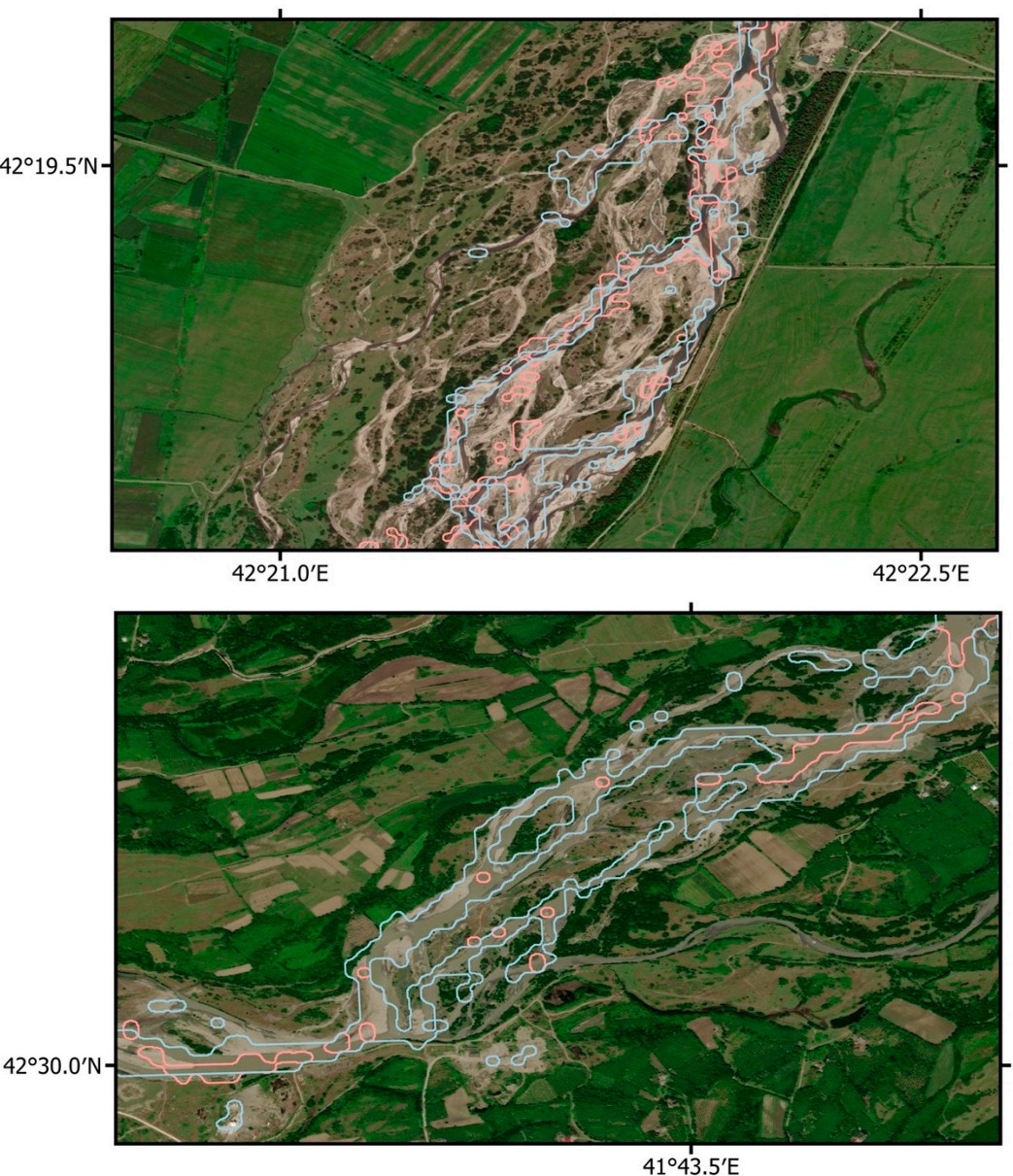

**Figure 5.** Two examples of the braided river system in Georgia. The rivers are broad and move around a fair amount, which is clear from the underlying satellite image. The OPC water data is shown in blue, with the JRC water data in pink. The OPC data picks up more of the smaller water channels.

### 4.5. Detection of Small Water Bodies and Irrigation Channels

Evaluating the performance of the two water datasets solely with statistical evaluations does not fully represent their ability to discriminate between the different sizes and types of water bodies present in the satellite image. Previous studies have already shown that the JRC water dataset is highly accurate in detecting large bodies of surface water but tends to omit smaller bodies of water, including water bodies that contain significant amounts of vegetation [32,43]. Examples of the omission errors for these types of water bodies are highlighted in Figures 6 and 7. Figure 6 shows the difference in performance between the two water detection methods in identifying small floodwater areas within agricultural plots. The OPC method can detect this water type, whereas the JRC method remains insensitive to it. In addition to the floodwater areas omitted by the JRC water map, small irrigation channels also go undetected. In contrast, the OPC method can delineate these water types, giving a more accurate representation of the surface water present in the scene.

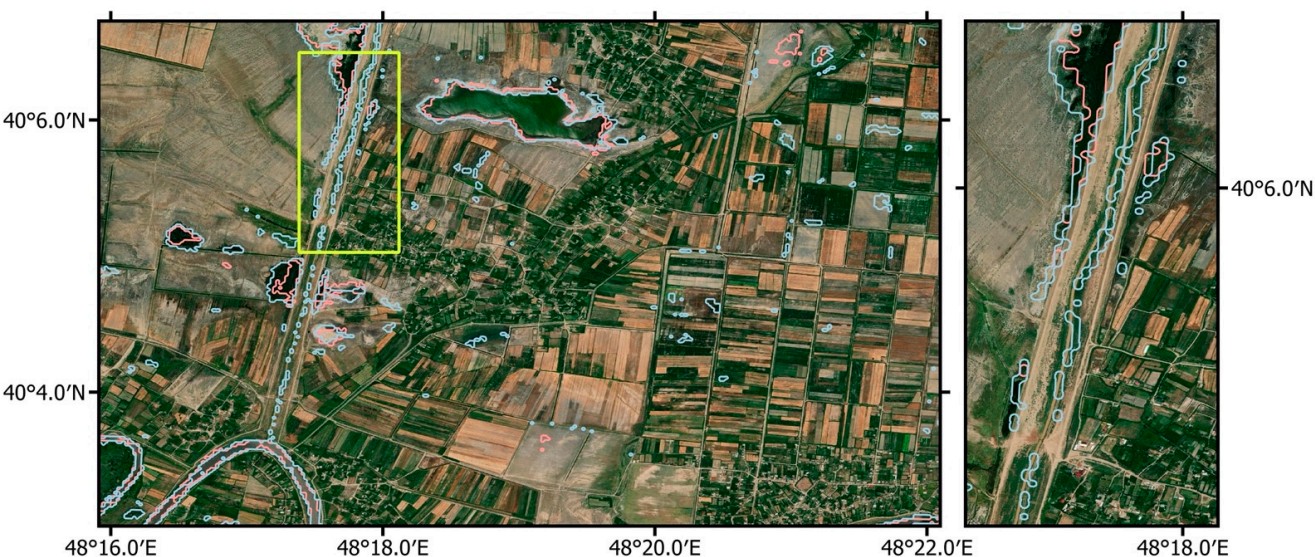

**Figure 6.** The difference in floodwater and canal detection between the JRC and the OPC Max Extent water maps. The water detected in the JRC Max Extent dataset is shown in pink, while the water detected in the OPC Max Extent map is shown in blue. At the bottom left, the river is detected in both datasets, while the small irrigation canal that branches off to the north is only found in the OPC dataset, easily seen in the smaller inset (yellow box) on the right. The larger water body to the east of the irrigation channel is found in both datasets, while the smaller water bodies to the left and right of the irrigation channel are visible in the OPC dataset, but not complete in the JRC data. The OPC areas on the agricultural fields in the east were verified as flooded fields at the time of the surface water recording.

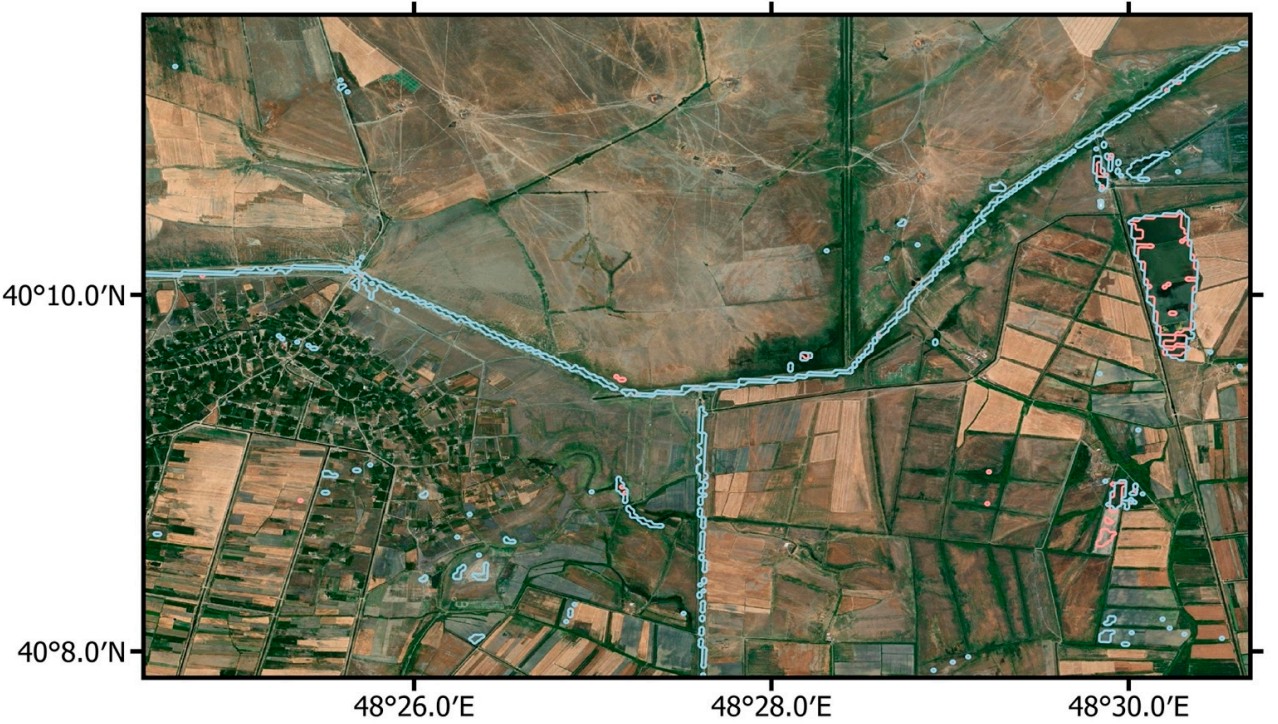

**Figure 7.** Comparison of the ability of the OPC and JRC methods to detect water in the Bash Shirvan Canal, northeast of Zəngənə, Azerbaijan. The canal water detected by the OPC Max Extent in blue, compared with the canal water detected by the JRC Max Extent in pink. The water body on the east is detected in both datasets, but the canal is only visible in the OPC dataset.

The difficulties in detecting surface water from the JRC dataset are not limited to small and vegetated bodies of water. Major water canals can also be a source of confusion in the

JRC data, as shown by the omission of the canal infrastructure in Azerbaijan (Figure 7). Here we can see that the JRC Max Extent water map almost completely ignores this section of the canal. Alternatively, the OPC method proves to be more reliable in detecting the water present in the canal, giving a more accurate representation of the surface water present in the scene. The omission of shallow, vegetated, and unstable water bodies can result in significant underestimation of surface water area and possibly affect the conclusions of studies that use this dataset in their model. Some of the very small irrigation channels are also not accurately detected in the OPC data.

## 5. Discussion

### 5.1. Challenges in Water Detection

Olofsson et al. (2013) found that a map can be highly accurate while still having low accuracy in detecting individual classes due to bias, suggesting that commonly-used accuracy metrics such as overall accuracy and the kappa coefficient do not take full advantage of the accuracy assessment data [56]. They recommend including user's, producer's, and overall accuracies, along with an area-adjusted map classification error, and generating confidence intervals for the adjusted area estimates [56].

We evaluated the user's, producer's, and overall accuracies, in addition to the adjusted accuracy from the estimated proportion of the area. We applied these accuracy metrics to the Max Extent water maps from the JRC and OPC methods for 2019. In comparing the overall accuracy between the two Max Extent water maps, we see that both datasets are highly accurate when evaluating both classes together (86.7% JRC, 89.2% OPC). Despite retaining high overall accuracy across the study area, assessing the accuracy of the two water maps by class provides insight into the effectiveness of the datasets in relation to water classification. When evaluating the estimated proportion producer's accuracy for water, the accuracy drops for the JRC and OPC Max Extent water maps (11.5%/15.7%), displaying a substantial underestimation of the surface water present. Comparing the path/row sets (Figure 4), we observe a significant reduction in the producer's accuracy for detecting the water class. The overall accuracy of the water maps is misleading in evaluating performance because each class is weighted equally in the calculation directly from the sample count error matrix [56,59]. In this region inland water is a rare class, making up around 2% of the total land cover in both water maps. The unweighted accuracy of both water datasets suggests that the water detection methods accurately detect water. However, a proportionally weighted accuracy assessment shows us that the water maps accurately distinguish non-water land cover. By weighting the error matrix by the proportion of estimated area, we observe a very low producer accuracy, suggesting that a large portion of water is omitted from both water maps, contradicting the viability of traditional unweighted accuracy assessments. Such a loss in performance between the JRC and OPC in detecting water shows that it is necessary to use an unbiased estimator of the area's proportion to properly weight each class in a sample count error matrix to avoid bias and strengthen land cover accuracy assessments.

### 5.2. Vegetated and Very Small Waterbodies

Detection of small, dynamic, vegetated surface water bodies is essential for monitoring the risk of mosquito-borne illnesses [60]. We find that the JRC dataset particularly struggles to detect the shallow, vegetated water bodies which are typical in the agricultural areas of the South Caucasus (Figures 8 and 9). The presence of vegetation in water can change the spectral signature, depending on the distribution of its subpixel components [33]. The OPC method proves to be more resilient to the spectral deviations of pixels that contain water with vegetation, in addition to the improved identification of water bodies that have unusual configurations, such as irrigation canals.

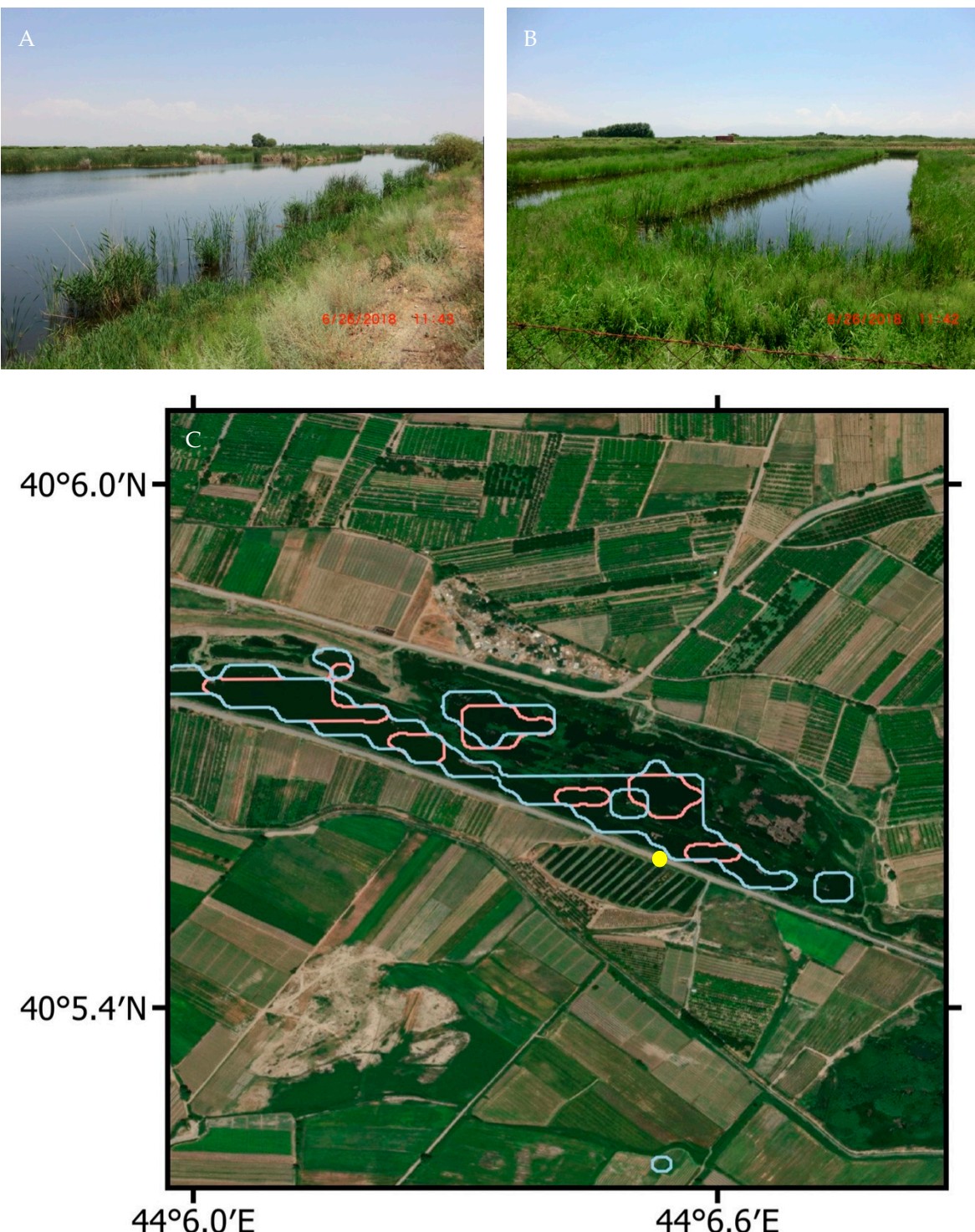

**Figure 8.** The two photographs (**A** & **B**) are taken looking north (**A**) and south (**B**) from the yellow dot in the middle of the image, showing an agricultural area of southern Armenia (**C**). The larger water body to the north that is partially vegetated is largely captured by the OPC data, with fewer areas of water detected in the JRC data. The aquaculture water bodies in the photograph looking south are too small to be detected in either dataset.

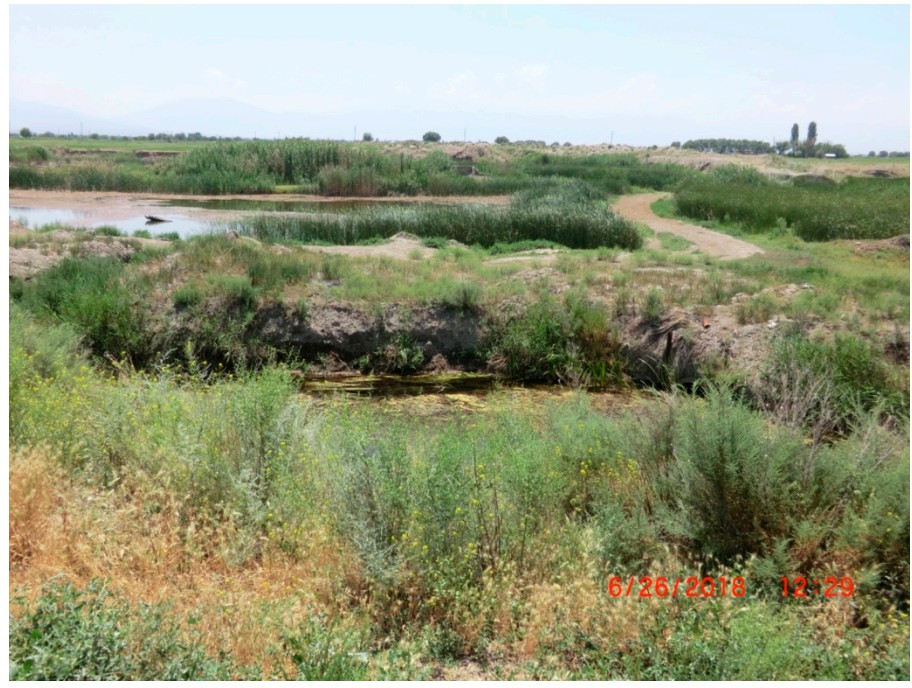

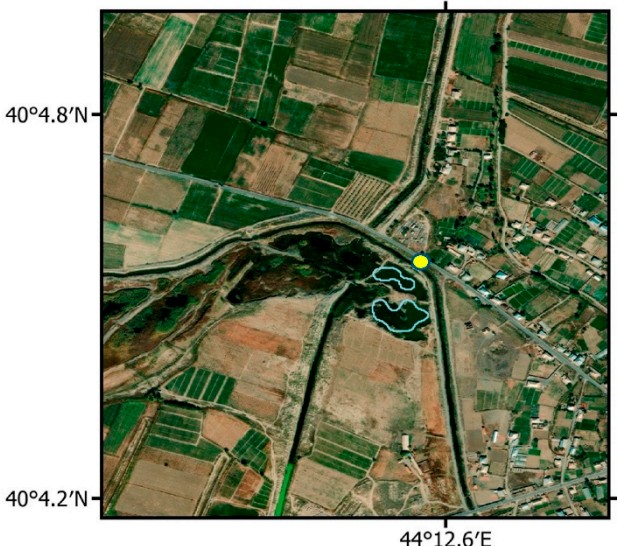

**Figure 9.** This photograph (**top**) is looking south from the yellow dot on the image (**bottom**), showing southern Armenia. The irrigation canal closest to the photographer is not visible in either the OPC or the JRC dataset. However, the larger marshy water bodies toward the top of the photograph are found in the OPC data but not in the JRC data.

We made field observations in the summer of 2018 at many of the large marshy areas found in southern Armenia. Figure 9 shows how close these marshy areas are to the surrounding farmlands and villages. This is one of the primary areas of malaria infection in the early 2000s in Armenia. These small, vegetated water bodies are difficult to detect in both datasets. Water detection methods must continue to develop improved sensitivity to these areas, perhaps using a broader range of openly available data and including active sensors such as Sentinel-1 and cloud computing tools [60,61].

## 6. Conclusions

Previous work has demonstrated the utility of satellite earth observations for monitoring environmental regions with malaria transmission risk; however, much of that work was focused on tropical areas of Africa, Asia, and Central and South America [60,62]. Here

we look at a temperate region spanning the boundary of Europe and Asia, which is also vulnerable to mosquito-borne illnesses, and we demonstrate the use of a global surface water dataset to create improved regional data for detection of the type of surface water that indicates potential mosquito breeding habitats.

Global land cover datasets are becoming essential to land cover detection in remote sensing communities, with several products freely available to users [57,63–65]. These datasets are especially useful as reference data in areas with little opportunity to collect sample data [46]. A study focused on urban regions improved urban area maps using the European Space Agency's GlobCover product to train a classifier based on a multinomial logistic regression [46]. Global land cover datasets have given us great insight into land cover presence, distribution, and temporal behavior [57,64,65]. We can also use these global datasets to train classifiers for regional studies [45].

Here, we displayed that applying a logistic regression using the JRC water product can improve the performance of the original water dataset retroactively. We previously demonstrated that the JRC tends to underestimate the area of water present [32]. The underestimation of surface water reveals that the JRC dataset is conservative in its ability to delineate water from other land cover types, omitting small water bodies in the final product. We used this conservative nature of the JRC water dataset to our advantage; as we can trust that water is present in locations where it is indicated by the JRC dataset, we used the JRC dataset to train a logistic regression water model and optimal probability cut-off to improve regional surface water maps.

We demonstrated our method by applying it across the entire Southern Caucasus region for May to October 2019 using training points generated from the JRC monthly history and yearly max extent water datasets. Comparing the JRC and OPC max extent water maps, we found that both are highly accurate, with an overall accuracy of 86.7% and 89.2%, respectively, when applied over the Caucasus region. The fact that this global JRC dataset has such high accuracy is a tremendous feat by itself. We demonstrated that our OPC method trained on the highly accurate JRC water datasets has increased sensitivity to small water bodies, detecting 392 km$^2$ more water than the JRC max extent water map for the entire year. We prove the viability of using existing global datasets to train a model and improve accuracy, giving a better representation of regional total surface water area.

**Author Contributions:** Conceptualization, J.W. and K.M.d.B.; methodology, J.W.; software, J.W., K.M.d.B. and B.C.O.; validation, J.W.; formal analysis, J.W.; investigation, J.W.; writing—original draft preparation, J.W.; writing—review and editing, J.W., K.M.d.B., J.K. and B.C.O.; visualization, J.W., K.M.d.B. and B.C.O.; supervision, K.M.d.B.; project administration, K.M.d.B.; funding acquisition, K.M.d.B. All authors have read and agreed to the published version of the manuscript.

**Funding:** This research was supported by the NASA Land-Cover and Land-Use Change project entitled "Land use patterns and political instability as predictors for the re-emergence of malaria in the Caucasus" to KMB. Project number: 16-LCLUC16-2-0017.

**Data Availability Statement:** We have made the validation points and the R code to run our Google Earth Engine analysis [66] freely available on GitHub (https://github.com/landchange/Caucasus-water, 24 November 2021).

**Acknowledgments:** We thank three anonymous reviewers for their careful comments on our paper. We thank Ani Melkonyan-Gottschalk for her help during our field visits.

**Conflicts of Interest:** The authors declare no conflict of interest.

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
