# Peer review of "Application of Spectral Index-Based Logistic Regression to Detect Inland Water in the South Caucasus"

_remotesensing, doi:10.3390/rs13245099_

Round 1
Reviewer 1 Report
The manuscript entitled “Application of spectral index-based logistic regression to detect inland water in the South Caucasus” by James Worden, Kirsten M. de Beurs, Jennifer Koch and Braden C. Owsley proposes a new remote sensing algorithm for the estimation of inland water surface especially for small water bodies.
Here are my major comments per section.
Introduction – I think that authors need to improve the introduction section, first by justifying the selection of the period and year of this analysis, and second to work more on the need and relevance for downscaling a global product – especially for your study site. I noticed the lack of some supporting references in the text, and it would be good to have a paragraph comparing downscaling studies instead of studies that compare downscaling products to a global scale product, which I think it is unfair to compare a local tuned product with a global one and request the same accuracy.
Background and Study Region – This section is very confusing; my recommendation would be to re-write it.
Data and Methods – The methods are not well described and some of the procedures need some references. Authors should justify the use of the selected procedures. Additionally, the comparison between Landsat 5 and Landsat 8 images you need to run a radiometric calibration between sensors in order to correct for it, as well as consider the difference in the spectral bands.
Results – I think results should be nice to add some graphical results to visually interpret the performance of the product.
Discussion – I will the lack of discussing the results from Figure 4. I also think authors should discuss more about the limitations of the proposed method, for example the how significant is the improvement, the calibration/normalization of the Landsat 5 and Landsat 8 images, the causes for different performances in different images and a comparison to another regional product which does not use the JCR product, other methodological problems which was not described or justified.
Specific comments:
L35-L37 – I would recommend authors to add at least one reference per method that you described instead of having one reference for all in the end of the sentence.
L40-L41 – Need a reference to support this statement.
L41-L43 – Need a reference to support this statement.
L49 – Maybe it would be important to explain what type of in situ data you need (i.e. GPS coordinates of water, or a vector file of the water extent made by GPS). Just adding in situ data is too vague.
L49 – I would not say that Landsat is commonly used for training data, especially because we usually use Landsat data to develop products and use high spatial resolution data to create training datasets. It is not good to train a Landsat-based product with Landsat data, it should be more accurate.
L56 – I think it is important to acknowledge that global products should not be compared to regional or local products once the quality of a downscaling product is always better than a global product. I feel it is a bit unfair do compare a global to a local product.
L82 – Why this period? Why not the entire year to capture the season dynamics? I think authors should justify the selected period, as well as for the year of 2019, was it a normal year in terms of climatology? Maybe it was a wet year which could change the water surface size.
L88-L89 – Need a reference to support this statement.
L89 – Strange change of topic, without any explanation.
L102 – I think you should start with this, explaining that malaria is transmitted due to a vector-borne mosquito and then explain the area and the malaria outbreaks. These two paragraphs are very confusing and I would recommend authors to re-write them, maybe a re-ordering of the sentences may help as well.
L108 – Need a reference to support this statement.
L108 – Maybe it would be good to define what is a good breeding area, for example, what type of environmental conditions is needed (warm temperatures x cold temperatures).
L111 – Since you do not have a subsection 2.2 I do not see the reason to have a 2.1.
L121 – How can you support that the area is recovering? How much % is recovering? Need some references to support it.
L126 – Figure 1 – Not sure why this Figure is needed.
L129 – Should be 3.1
L133 – Why did you select 30%? Was it based in a previous work?
L134 – Again, it is important to explain why this period was selected.
L138 – I think there are some sentences missing here. After this sentence there is only the Figure and this sentenced ended with an “and…”
L142 – Why only in the area where tiles were overlapping? This is not clear, usually, the best is to use the center of the image not in the edge of the image.
L154 – Why did you use the MNDWI? There are several other indices for water surface extraction, why this one? I think authors need to justify their choices.
L161 – Was the image from Google Earth also collected in the same year of the images used?
L169 – I think this process is tendentious, maybe you should add some references of other studies using this.
L175 – This is very subjective, maybe you need to clarify what is “unidentifiable in the Google Earth images” or at least provide a guide (in a figure) what was considered for each class.
L191 – Doing this you should acknowledge the uncertainty of the JRC product
L194 – Please increase the resolution of this image.
L233 – Did you perform a radiometric calibration to compare Landsat 5 and Landsat 8 images? Because without a radiometric calibration between the two sensors, it is not possible to compare both products, especially when the spectral bands changed a little bit.
L265 – How significant is this change in accuracy?
L270 – it should be 4.2
L305 – Need a reference to support this statement.
L307 – How can you access and quantify this visual inspection? Maybe it would be better to visually show how this was done.
L322 – How much the change in the spectral resolution from Landsat 5 to Landsat 8 changes the MNDWI?
L347 – It would be easier to show both contour lines for the aquatic systems in the same image.
L387 – Maybe it would be better to test it on Level 1 data without the atmospheric correction.
Reviewer 2 Report
A good work, which merit publication, with minor revisions. The reference section need to be improved and include some more case studies from Meditteranean, giving a more international attitute. The proposed works, are mentioned below:
Ighalo J.O., Adeniyi A.G., Marques G., 2021. Artificial Intelligence for Surface Water Quality Monitoring and Assessment: A Systematic Literature Analysis. Modeling Earth Systems and Environment, Vol 7, pp 669–681
Psilovikos A. & Elhag M., 2013. Forecasting of Remotely Sensed Daily Evapotranspiration Data over Nile Delta Region, Egypt. Water Resources Management, Vol 27, pp 4115–4130, DOI 10.1007/s11269-013-0368-2.
Diego Gómez, Pablo Salvador, Julia Sanza José & Luis Casanova, 2021. A New Approach to Monitor Water Quality in the Menor sea (Spain) using Satellite Data and Machine Learning Methods. Environmental Pollution, Volume 286, 117489.
Reviewer 3 Report
This study developed a new surface water detecting method based on previous JCR water dataset using the optimal probability cutoff (OPC) logistic regression model. The results show that the new method performs well in capturing small area water, which is significant in the study area due to the water-mosquito-malaria relationship. However, some describes in this paper are still vague:
- At the end of the introduction, the objective of this study is vague. It seems to improve the detective capability of the band ratio water index, especially for smaller water bodies. In that case, the case study of Lake Sevan is mismatched, because the lake could not represent the small area surface water.
- Figure 1 is less informative, neither location nor size of the lake. By contrast, photos of vegetated water bodies, irrigation canals, and braided river systems mentioned in Figure 7, Figure 8 and discussion parts are more needed.
- Time series are mentioned many times, but it is still confusing why time series analysis is employed and how seasonal changes influence the result. For example, in line 309, authors say the water in path 168 is unstable due to vegetation growth, which changes during a whole year. This kind of change could be detected by monthly products of JCR. Therefore, maybe Figure 5 could be illustrated on a monthly scale. What’s more, seasonal surface water is excluded in validation point selection, but included in training data selection. This also brings uncertainties in later analysis. Over all, please explain the reason why these temporal scales are used in corresponding analysis.
- Two methods conducted in the study areas, the three countries, to detect water bodies. Although related statistical features are listed, the distribution of surface water is unknown. Surface water maps in the whole study area need to be added, which is also needed in analysis of lines 300-316.
- In figure 6,7 and 8, the small area of water is still unclear on Google Earth Image. Multiscale maps and photos could be combined here to provide a more clear description.
- This study used the validation point from Google Earth images. While validation based on other higher-resolution images, such as Sentinel and QuickBirds, may provide a more convincing comparison.
- In lines 165-172, although the bias from missing data is offset by adding 500 additional water validation points where showed a divergence between the JRC data and the data generated in this study, the quality of JCR in the study area should be assessed because the highly relying on JCR datasets, which is still facing a large proportion of missing data in some areas.
Round 2
Reviewer 1 Report
I would like to thank the authors for providing this revised version I appreciated that many of my comments were addressed, however, as stated in my previous report, I do not think that authors improved the discussion and the methodology is still confusing.
I am not sure why authors have a section (the Background section) with only one paragraph in it. Maybe it would be better to incorporate this paragraph in the introduction. Another option is to write a proper background section describing the state of the art. I understand that authors suggest that not a lot of studies were conducted in your study site, however, several studies trying to downscale global products were performed worldwide and those are the studies in which this study should compare. Are you getting similar results to the ones who attempted it in other regions? What kind of methods are they using? I do not think it is an explanation the fact that there are no other studies in this area, because here you are presenting a methodology which could be used in different parts of the world, therefore, authors should compare to other methodologies not only to the ones applied to the same study site.
In the previous version, one of the key problems was the radiometric calibration between Landsat 5 TM and Landsat 8 OLI which authors decided to remove the Landsat 5 TM data instead of performing a calibration. I was disappointed with this decision, because it is important to look for the harmonization of datasets, I know it would give more work but it would improve the interest for the community.
The discussion is still not satisfactory, as mentioned before, there are several studies and methodologies applied to downscale a global product in other regions and those should be compared for a discussion about your proposed method.
Minor comments:
For the new figures, please add a scale and a legend within the figure (not only an explanation in the caption.
Reviewer 3 Report
The figure 1 is not standard drawing. The legend and scale need to be added. The position of the study are had better to be shown in the bigger area.
